# Visual Intelligence in Precision Agriculture: Exploring Plant Disease Detection via Efficient Vision Transformers

**DOI:** 10.3390/s23156949

**Published:** 2023-08-04

**Authors:** Sana Parez, Naqqash Dilshad, Norah Saleh Alghamdi, Turki M. Alanazi, Jong Weon Lee

**Affiliations:** 1Department of Software, Sejong University, Seoul 05006, Republic of Korea; parez.sana@gmail.com; 2Department of Convergence Engineering for Intelligent Drone, Sejong University, Seoul 05006, Republic of Korea; dilshad.naqqash@gmail.com; 3Department of Computer Sciences, College of Computer and Information Sciences, Princess Nourah bint Abdulrahman University, P.O. Box 84428, Riyadh 11671, Saudi Arabia; nosalghamdi@pnu.edu.sa; 4Department of Electrical Engineering, College of Engineering, Jouf University, Sakaka 72388, Saudi Arabia; tmanazi@ju.edu.sa

**Keywords:** agriculture monitoring, deep learning, embedded vision, Internet of Things (IoT), image classification, plant disease detection, vision transformers, precision agriculture

## Abstract

In order for a country’s economy to grow, agricultural development is essential. Plant diseases, however, severely hamper crop growth rate and quality. In the absence of domain experts and with low contrast information, accurate identification of these diseases is very challenging and time-consuming. This leads to an agricultural management system in need of a method for automatically detecting disease at an early stage. As a consequence of dimensionality reduction, CNN-based models use pooling layers, which results in the loss of vital information, including the precise location of the most prominent features. In response to these challenges, we propose a fine-tuned technique, *GreenViT*, for detecting plant infections and diseases based on Vision Transformers (ViTs). Similar to word embedding, we divide the input image into smaller blocks or patches and feed these to the ViT sequentially. Our approach leverages the strengths of ViTs in order to overcome the problems associated with CNN-based models. Experiments on widely used benchmark datasets were conducted to evaluate the proposed *GreenViT* performance. Based on the obtained experimental outcomes, the proposed technique outperforms state-of-the-art (SOTA) CNN models for detecting plant diseases.

## 1. Introduction

Over the past few decades, agriculture has emerged as the primary source of income for several countries, significantly contributing to the global economy. As per the World Bank report of 2018, agriculture engaged over a billion population, representing 28.5% of the total labor force, and amounted to about 10 million tons of food a day [1]. Although, the total potential of agriculture is prone to plant infections and diseases, food security is compromised by such infections. Major food crops, such as rice, wheat, potatoes, soybeans and maize can suffer losses of 10% to 40% due to plant viruses [2]. Addressing these challenges necessitates frequent inspection of disease symptoms, often inefficient and time consuming, particularly for huge crop fields. In order to achieve precision agriculture, plant infections must be detected effectively. Research groups have been motivated to work on Machine Learning (ML) as a result of the proliferation of this field to explore its potential in automating the detection of plant diseases by analyzing images obtained from fields. To identify diseases, these groups analyze the images and extract significant features. For example, in Ref. [3], Support Vector Machine (SVM) was applied after extracting image features using Scale Invariant Feature Transform (SIFT) to classify guava leaf diseases. Before applying SVM, Ref. [4] used a Statistical Gray Level Co-occurrence Matrix instead of SIFT. A variety of feature extraction methods have been utilized by other studies to analyze plant diseases in order to achieve remarkable results, including [5,6,7,8].

To handle fiddly data designs and stupendous training data, some researchers used the k-Nearest Neighbors (k-NN) classifier instead of SVM. It was possible to classify Cotton Grey Mildew disease based on local statistical input features and k-NN in [9]. There has also been use of the k-NN algorithm in [10] for classifying paddy leaves and [11] for classifying groundnut leaf diseases. However, every approach requires multiple steps in order to prepare the data for preprocessing and feature extraction. There is also no evidence that they are effective at classifying more than one class of data and that they are sensitive to predefined parameters, such as the kernel parameter k in SVM and the kernel parameter f in K-NN [12]. To address these issues, some researchers have turned to Deep Learning (DL) methods for improved crop infection and disease detection. The authors in [13] created a DL-based system called PlantVillage (PV), which can accurately identify 26 different plant diseases. In contrast to explicit feature extraction techniques, DL techniques automatically learn and extract relevant features from the input images. However, traditional ANN classifiers lose spatial information when converting 2D images to 1D vectors for classification, leading to increased computational complexity and storage requirements. The article in [14] describes a method for defining diseases of plants based on a new dataset called DRLI.

Agricultural applications of neural networks, specifically convolutional neural networks (CNNs), have proven successful in overcoming previous limitations with Deep Neural Networks (DNNs) [15]. For example, Ref. [16] used the MaskRCNN model with transfer learning to detect fusarium head blight disease in wheat, achieving an average accuracy of 92.01% on the intended test data, which included around 450 images. In a similar manner, a process was applied to analyze [17] and identify apple leaf diseases with 77.65%, 75.59% and 73.50% recognition accuracy using ResNet152, Inception V3 and MobileNet models. According to [18], the authors proposed using the PV dataset to generate a custom DCNN to classify cucumber infections that performed up to 94% accurately, using a pretrained AlexNet model. In [19], the authors developed a Custom-Net model for classifying pearl millet diseases using Raspberry Pi (RPi), achieving an accuracy of 98.78%. Additionally, DL models have been applied to detect further leaf infections, mostly established on the PV dataset. According to the mentioned methods, the accuracy of classification has been high, especially in the case of tomato leaf diseases, which obtained 97.49% results in [20], and with banana leaf diseases, which obtained 99.72% results in [21].

Moreover, AI models and ML techniques have been deployed on drones, which are unmanned aerial vehicles used to mitigate various malicious factors in agriculture, including lack of rain, nutrition abnormalities, infections, weeds and pests. Precision agriculture uses drones widely because they are affordable, have an extensive range of operation and are AI-compatible, according to [22]. Based on the IoT architecture used by [23], the researchers equipped a drone with an RPi 4 to classify plant infections early in crop production by using the IoT. As reported in [24], a drone-based fidelity farming apparatus is being used to detect affected areas in open-field tomato crops using algorithmic neural networks. Depending on the level of infection, precise pesticides could be sprayed on afflicted areas. In a similar way, the [25] project constructed an automated system to detect and spray chemicals on infected plants by combining high-quality cameras with disease detection models based on ResNet architecture.

While SOTA DL models can achieve high-performance results and are suitable for use with drones, they require significant computational resources for training. In contrast, ViT which avoids CNN and has similar performance to SOTA CNN models, is a promising alternative [26]. As a derivative of Transformer, ViT employs a self-attention mechanism that determines a global reference pixel-by-pixel during training. Each given image is split into equal patches and each patch is embedded with its position. After learning from patches, the self-attention block can accurately represent them for vision tasks. In the last layers of the ViT model, the cosine similarity between patch representations increases significantly, suggesting that increasing the layers does not enhance model performance [27]. The memory requirements of ViT make handling high-resolution images difficult due to their four times length requirement.

Several studies have aimed to overcome the limitations of Transformer-based models, and in general, it falls into two categories: hybrid models and pure-Transformer enhancement. Hybrid models combine the strengths of CNN and Transformer to improve performance. A model based on CNN called Ghost-Enlightened Transformer, for example, was proposed by [28] to construct intermediate feature maps. In the next step, the self-attention mechanism is used to convert those maps into deep semantic features. Based on 12,615 images collected by the author, this model achieved 98.14% accuracy. A similar system is outlined in [29]. As is PlantXViT, it incorporates a VGG16 network, a transmission block and an encoder layer called Transformer. The VGG16 and inception block provide better capture of local image features than SOTA CNN models currently available. Furthermore, multiple studies have incorporated CNN layers into Transformer architectures to amplify the capability of extraction of most prominent features [30,31,32]. As a result of this approach, the model becomes more accurate because it is able to learn local features through the CNN architecture, but the training and inference times are significantly extended and the memory used is huge.

In contrast, pure-Transformer enhancement variants operate primarily based on optimizing the self-attention mechanism to improve performance. Based on shifted windows, the Swin Transformer, for example, calculates local attention efficiently while maintaining connections across windows [33]. Additionally, Ref. [34] developed k-NN attention, which determines the attention matrix based on the top-k related tokens found in the keys, thereby reducing training time. In RegionViT, local self-attention is employed to retain global information through a regional-to-local concept [35]. Several studies have also proposed modifying the self-attention mechanism by using feature channels instead of tokens in the calculation of the self-attention matrix [36] and revamping the spatial attention mechanism to include small-distance, large-distance and all-inclusive information [37]. It involves optimizing the attention matrix calculation process in order to decrease the model’s complexity while maintaining the global connection. There are, however, some studies that maintain the original architecture of the self-attention mechanism, leading to a huge number of trainable parameters in each self-attention head in comparison with previous studies. Thus, existing Transformer-based models retain their complexity while being larger. Transformer-based models have these limitations, which hinder their application to intelligent edge applications, such as drones and single-board computers, where resources are limited. We designed the models so that they could be deployed and operated on products that have limited resources, with the aim of minimizing transmission latency and network bandwidth consumption [38]. In summary, this study made the following contributions:Plant disease detection is now significantly improved using CNN-based models, based on the latest research findings. However, the particular models exhibit limitations such as translation invariance, locality sensitivity and a lack of global image comprehension. To address these shortcomings inherent in CNN-based approaches, this study introduces a new approach utilizing a Vision Transformer-based model for improved and effective plant disease classification.Drawing inspiration from the Vision Transformer (ViT) proposed by Alexey Dosovitskiy et al. [26], we conducted training and fine-tuning of the ViT model, specifically for fire detection, resulting in notable advancements surpassing the SOTA CNN models. By improving the architecture of the ViT model, it has been possible to reduce the number of learning parameters from 85 million to 21.65 million as a result of the fine-tuning process, which has resulted in an increase in the accuracy of the model at the same time.The proposed *GreenViT* model exhibits exceptional accuracy and effectively reduced the occurrence of false alarms. Consequently, the developed system proves to be highly suitable for accurate plant disease detection, ultimately mitigating the risks associated with food scarcity and security.

This paper is further divided into the following sections: In Section 2, the proposed methodology is presented, outlining the key steps and techniques employed in the study. Section 3 provides a brief description of the experimental results obtained from the conducted experiments. Finally, in Section 4, the paper is concluded, summarizing the main findings, contributions and prospective approach for future work.

## 2. Material and Methods

This section begins by introducing the experimental dataset used in the study. Subsequently, the plant disease detection model, named *GreenViT*, is presented. It is necessary to review the experimental environment, as well as evaluation metrics in order to evaluate the performance of the model.

### 2.1. Datasets

To gauge the effectiveness of the proposed model, the study utilized two popular standard datasets, namely PV and DRLI. Furthermore, to test the model’s resilience, a new dataset called PC dataset was utilized, which was created by integrating both datasets. The combined datasets’ statistics are listed in Table 1 while the comprehensive details are provided below.

#### 2.1.1. Plant Village

A feasibility study was conducted to assess the effectiveness of the newly presented *GreenViT* method; the authors conducted experiments on two well-known benchmark datasets PV and DRLI. The PV dataset has been widely utilized in previous studies due to its large size, public availability and free access to data on crop leaf disease classification. To validate the classification accuracy of the employed approach, the authors carried out several experiments on this dataset, which comprises images of plants with various types of diseases. The dataset contains a total of 54,303 images from 14 plant species, which are categorized into 38 classes. Of these, 26 classes correspond to infected plants, while 12 belong to healthy plants. The dataset includes images of plants such as tomatoes, strawberries, grapes and oranges. In addition to variations in color, size and lighting, the dataset features image distortions such as noise, blurring and color variations, making it a challenging dataset for detecting and categorizing affected plant leaf regions.

#### 2.1.2. Data Repository of Leaf Images

The intricate interaction between plants and their surroundings leads to the production of various substances that enhance the environment and help in controlling greenhouse gases and climate change. However, in the past, humans have ruthlessly exterminated many plant species, resulting in the loss of biodiversity and further exacerbating climate change. To address this, the identification, detection and diagnosis of plant diseases have become crucial. In this dataset, the authors have chosen twelve plant species, including guava, arjun, mango, alstonia, bael, scholaris, jatropha, jamun, pomegranate, basil and lemon. The leaves of these plants were photographed in both healthy and infected states and were divided into two categories: healthy and infected. The entire dataset contains approximately 4503 photos, with 2278 healthy leaves and 2225 diseased leaves, taken from March to May 2019 at the University of Shri Mata Vaishno Devi in Katra. The dataset was divided into 22 subject groups based on plant species, and the photographs were captured in an enclosed space using a Nikon D5300 camera (Nikon, Tokyo, Japan) with an 18–55 mm lens and sRGB color representation. The photos were taken with 1000 ISO and without flash, resulting in a single JPEG photo in 0.58 s per frame and a RAW + JPEG photo in 0.63 s per frame.

#### 2.1.3. Plant Composite

In order to evaluate the robustness of the proposed *GreenViT* model, the authors conducted an experiment using a combination of publicly available datasets: PV and the DRLI. By merging these datasets, a new and more diverse dataset was created, which posed greater challenges for the model. The composite dataset consists of a total of 58,807 images, making it 7.6% larger than PV and 92.3% larger than the data repository of leaf images. This increased size and diversity of plant species within the dataset necessitated a meticulous training process for the model. As a result, the model demonstrated improved the generalization ability and enhanced the reliability for real-time plant disease detection scenarios by providing a visual representation.

### 2.2. The Proposed GreenViT Plant Disease Detection Method

The proposed framework has been thoroughly outlined in this section. A Transformer model forms the foundation for our framework. Currently, the Transformer model is widely regarded as the SOTA in handling sequential data processing, particularly in Natural Language Processing (NLP) tasks for instance speech recognition, language modeling and machine translation. The Transformer architecture, introduced by [40], revolves around an encoder–decoder module that facilitates the rearranging and incorporating of a given sequence of elements into a new sequence. The primary objective behind the development of Transformers was to enable parallel processing of data. The purpose of this study is to evaluate the performance of the ViT model in predicting plant diseases. As depicted in Figure 1, the ViT architecture is employed, which takes an input image with dimensions of 72×72 pixels. Initially, the input image is divided into patches, and the number of patches utilized depends on the specific scenario being addressed. In this study, the input image is converted into six image patches. To accommodate 2D images with height (*H*), width (*W*) and (*C*) channels, the image, denoted as $X\in {ℜ}^{\left(H\times W\times C\right)}$ is reshaped into a sequence structure resembling word embedding. This transformed representation is then used as input to the transformer network, which processes the 2D patches (*P*) $X_{P}\in {ℜ}^{N(P^{2},C)}$. This is a representation of the actual image $X_{P}$, and the resolution of the patches is characterized by (*P*, *P*). The most functional length of the sequence for the transformer is determined via $N=HW/P^{2}$. In the transformer network, these patches are treated in a similar manner as tokens in NLP. In each layer of the transformer, a fixed width is maintained, and a trainable linear projection is applied to map each vectorized patch to the model dimension *D*. The resulting outputs are referred to as patch embedding. The ViT model incorporates three main components: the embedding layer, the encoder layer and the classifier layer. These components will be discussed in detail as follows:

#### 2.2.1. Embedding Layer

Transform models treat patches individually as tokens and map them to higher dimensions through learnable linear projections. These embedded projections are then combined with a learnable class token $U_{Class}$ that plays a crucial role in the classification process. To preserve the positional information and to retain the spatial positioning of the patches, positional embedding $E_{Position}$ is employed. Each patch in the image can be located precisely based on these positional embedding. The patch concatenated with the token $Y_{0}$ is represented by the following Equation (Equation 1):(1)$$Y_{0}=\left(U_{Class};X_{P}^{1}E;X_{P}^{2}E;\ldots ;X_{P}^{n}E\right)$$

This equation captures the fusion of the class token $U_{Class}$ with the encoded patches to form the final input representation for further processing in the model.

#### 2.2.2. Encoding Layer

In this particular step, the transformer encoder plays a crucial role in processing a sequence of embedded patches, denoted as $Y_{0}$. The ViT utilizes a set of *L* encoder blocks, which are further subdivided into two distinct sub-components: Multi-Head Self-Attention (MHSA) and the Multi-Layer Perceptron (MLP). The MHSA block serves as a pivotal component within the encoder block, incorporating self-attention and concatenation layers. Specifically, given an input x=x1,x2,…,xn, an attention operation is performed with the transformer on a set of queries *Q* using all available keys *K* and values *V*. This process is represented in Equation (Equation 2).
(2)$$\mathrm{Attention}\left(Q,K,V\right)=\mathrm{SoftMax}\left(\frac{QK^{T}}{\sqrt{D}}\right)V$$

In Equation (Equation 2), the weight matrices $W^{Q}$, $W^{K}$ and $W^{V}$ are trainable parameters that determine the importance or weight assigned to the value, query and key, respectively. The process involves calculating the dot product of the queries *Q* across all keys *K*, scaling it by the square root of *D* and applying a SoftMax classifier for classification. The transformer executes multiple parallel iterations of scaled dot product attention using different weights, known as attention heads. The outputs of these attention heads are then merged together to calculate the end result, as listed in the Equation (Equation 3).
(3)$$\mathrm{MHSA}\left(Q,K,V\right)=\mathrm{Concatenate}\left({\mathrm{Attention}}^{1},{\mathrm{Attention}}^{2},\ldots ,{\mathrm{Attention}}^{n}\right)W^{0}$$

In Equation (Equation 3), $W_{i}^{Q}$, $W_{i}^{K}$, $W_{i}^{V}$ and $W^{0}$ refer to the trainable parameter matrices. The final output at the *I*th layer of the MHSA block is formulated in Equation (Equation 4).
(4)$$Z_{l}^{\prime }=\mathrm{MHSA}\left(LN\left(Z_{l-1}\right)\right)+Z_{l-1},where\;l=1,2,3,\ldots ,L$$

The MLP comprises two fully connected layers, which are connected sequentially. Following the fully connected layers, the ReLU activation function is applied. The output of the MLP is provided in Equation (Equation 5).
(5)$$Z_{l}=\mathrm{MLP}\left(LN\left(Z_{1}^{\prime }\right)\right)+Z_{1}^{\prime },where\;l=1,2,3,\ldots ,L$$

#### 2.2.3. Classification Layer

In the given sequence, the very initial entity $Z_{l}^{0}$ is extracted and passed on to an external head classifier responsible for predicting the last layer of the encoder. The head classifier performs classification by assigning the input to one of two corresponding class labels: “Healthy” or “Infected”. The formulation for this classification process is provided below in Equation (Equation 6).
(6)$$y=LN(Z_{l}^{0})$$

Alexey Dosovitskiy et al. [26] proposed three fundamental versions of the ViT, namely ViT-Base, ViT-Large and ViT-Huge. In each version, the number of encoders, hidden dimensions, attention heads and classifiers differ. The ViT-Base variant is trained using a patch size of 16×16, employing 12 layers in the encoder, a hidden size of 768 and 12 attention heads. On the other hand, the ViT-Large and ViT-Huge versions are computationally more demanding. For a detailed overview of the specifications for each version, please refer to Table 2.

During the experiments, the ViT-Base model is fine-tuned with specific configurations. The projection dimensions, number of heads, transformer layers and MLP head units are set to 64, 4, 8 and 1024, respectively. Following the MLP heads, a SoftMax classifier is employed for classification, distinguishing between two classes: “Healthy” and “Infected”. The tweaking process of the ViT-Base method successfully reduces the total number of learning parameters without compromising the overall performance.

## 3. Experimental Results

This section delves into the evaluation and assessment metrics, and graphical outcomes. We begin by describing the experimental setup and performance metrics. Then, we discuss the evaluated results. All models, including our proposed *GreenViT*, underwent training for a total of 10 epochs, employing a low learning rate to ensure the retention of previously acquired knowledge. The pre-trained model continually updated its learning parameters to optimize performance on the designated dataset. After obtaining the results, each model underwent retraining using its default input size of 224×224 while the proposed *GreenViT* utilized 72×72, also employing a batch size of 32. The Adam optimizer was utilized with a learning rate of $1\times 10^{-4}$ and momentum of 0.9. The experiments were conducted on an NVIDIA GeForce RTX 3090 Graphical Processing Unit (GPU) that has 24 GB on-chip memory, equipped with 64 GB of onboard memory (Nvidia Corporation, Santa Clara, CA, USA). The single-precision floating-point computing capability of the GPU can achieve a peak performance of 36 TFLOPS. For implementation, we utilized the Keras DL framework with TensorFlow 2.9.1 serving as the backend.

### 3.1. Evaluation Metrics

The proposed *GreenViT* model was assessed based on various evaluation metrics, such as precision, recall, F1-score and accuracy, where TP represents True Positive, TN represents True Negative, FP represents False Positive and FN depicts False Negative.
(7)$$\mathrm{Accuracy}=\left(\frac{\mathrm{TP}\;+\;\mathrm{FN}}{\mathrm{TP}+\mathrm{TN}+\mathrm{FP}+\mathrm{FN}}\right),$$
(8)$$\mathrm{Precision}=\left(\frac{\mathrm{TP}}{\mathrm{TP}+\mathrm{FP}}\right),$$
(9)$$\mathrm{Recall}=\left(\frac{\mathrm{TP}}{\mathrm{TP}+\mathrm{FP}}\right),$$
(10)$$F1\mathrm{-score}=2\times \left(\frac{\mathrm{Precision}\times \mathrm{Recall}}{\mathrm{Precission}\;+\;\mathrm{Recall}}\right).$$

### 3.2. Quantitative Results

This study conducted a comparison between the proposed *GreenViT* and various pre-trained CNN-based architectures for plant disease detection. The evaluation focused on the parameters, precision, recall, F1-score and accuracy. Among the models examined, such as VGG19, VGG16, EfficientNetB0, MobileNetV1 and MobileNetV3Small, most of them demonstrated similar performance. However, the base ViT performed the worst compared with the other models; on the other hand, the proposed *GreenViT* model achieved superior accuracies of 100%, 98% and 99% on all three datasets, while also exhibiting the lowest False Alarm Rate (FAR) compared with the other SOTA models. Notably, when comparing the proposed *GreenViT* with MobileNetV1, both models demonstrated computational efficiency, but the proposed *GreenViT* showcased low FAR and still outperformed all the included datasets. A detailed performance comparison of the employed models is listed in Table 3. It is evident that the pre-trained models achieve high performance with a low FAR. Nevertheless, the FAR remains elevated and necessitates improvement. Consequently, this research explores the refinement and pre-training of a CNN architecture, specifically *GreenViT*, with a focus on accuracy and reducing incorrect predictions. Following fine-tuning, *GreenViT* demonstrates the best performance among the other models, exhibiting fewer false predictions. Furthermore, the proposed *GreenViT* performance was evaluated employing 5-fold and 10-fold cross-validation on all the included datasets. The cross-validation accuracies show that our *GreenViT* maintains a competitive performance across all folds, even though there is a slight decrease in average test accuracy when the training samples in each fold are smaller compared with the whole dataset. This consistent performance demonstrates the robustness and reliability of *GreenViT*. Table 4 and Table 5 list a comprehensive overview of the 5-fold and 10-fold cross-validation accuracies for each dataset, including the average test accuracy across the 5 and 10 folds. These results reaffirm the effectiveness of our *GreenViT* in handling diverse datasets and its ability to yield consistent and promising results in real-world applications.

Figure 2 illustrates the confusion matrix of the *GreenViT* method trained on different benchmark datasets. The dark green diagonal corresponds to TP, while the saturation indicates accurate classifications. The proposed *GreenViT* demonstrates superior overall classification accuracy compared with the SOTA models, although there are some misclassifications within both categories. The training accuracy and loss graphs are visualized in Figure 3. The vertical axis represents accuracy and loss, while the horizontal axis represents the total number of epochs. It is evident from Figure 3 that *GreenViT* effectively detects plant diseases. As the number of training and validation iterations increases, the line graphs of training and validation accuracy change, as depicted in Figure 3a. The proposed *GreenViT* converges at seven epochs, achieving training and validation accuracies of 100%, 98% and 99% on the PV, DRLI and PC datasets, respectively. Similarly, the training and validation loss values change and decrease to 0.0 and 0.09, respectively, as depicted in Figure 3b. In addition, the suggested *GreenViT* is compared with the other pre-trained models in Table 3. The results indicate that the proposed *GreenViT* outperforms the other pre-trained models listed in Table 3.

### 3.3. Qualitative Results

We performed a visual analysis to determine the qualitative results of the proposed *GreenViT* model in distinguishing images with infection from those that are healthy plants based on class activation. The results, as shown in Figure 4, demonstrate the robustness of *GreenViT* in detecting diseased regions within a given input image. Figure 4 showcases the visual outcomes of the proposed *GreenViT* model for the samples obtained from the all three included datasets. The first row represents the input images from the PV dataset. The second row depicts the images from DRLI dataset. The third row contains images from the newly created PC dataset. All the samples are quite different from each other in type, size, geometry and color schema. The fourth row represents the ground truth (GT) labels which are the actual labels for each input image, while the last row shows the predicted label by the proposed *GreenViT* model. The infected images are highlighted in red, while the healthy samples are denoted by blue. The analysis depicted in Figure 4 offers compelling evidence of the remarkable capabilities of the proposed *GreenViT* model in accurately detecting infected and healthy regions of plant leaves. The visual representation vividly showcases the model’s proficiency in this aspect.

### 3.4. Time Complexity

In order to evaluate the effectiveness, performance and suitability for deployment of a DL model, it is crucial to conduct real-time assessments on various devices, including small edge device like the RPi 4 (Model B+), which incorporates a Central Processing Unit (CPU). The RPi 4B+ features a quad-core Cortex-A72 64-bit processor with 1.5 GHz, also comes with four GB of main memory. The specifications of the CPU analyzing the Frames Per Second (FPSs) of the proposed *GreenViT* model can be found in Section 3 of this paper. The established criterion for evaluating the model’s performance in optimal applications achieves an FPS of 30 or higher, which is considered optimal for real-world scenarios according to References [38,45]. To assess the model’s performance, the authors recorded a brief video of plants using a mobile phone. The FPSs obtained for the proposed *GreenViT* model when utilizing the RPi 4B+ and CPU are 0.34 and 22.19, respectively. Table 6 presents a comparison of the proposed *GreenViT* model’s FPS with that of several baseline models.

The experimental findings demonstrate the FPS achieved with different models, namely VGG19, VGG16, EfficientNetB0 and MobileNetV1, when employing the RPi 4B+ and CPU. For the VGG19 model, the obtained FPS values are 0.47 and 9.49. Similarly, the FPS for the VGG16 model is 0.62 and 11.09, while the EfficientNetB0 model achieves FPS values of 2.69 and 19.74. As for the MobileNetV1 and MobileNetV3Small model, the respective FPS values are 8.23, 22.96 and 7.43, 27.94. Comparing the inference speed of the ViT base variant and the proposed *GreenViT*, it becomes evident that the proposed model performs more favorably than the ViT Base and the modified *GreenViT* is a more suitable option for edge devices. This whole comparison supports the notion that the execution of the newly proposed *GreenViT* method is satisfactory. Therefore, the model exhibits a capability for real-time processing and operation.

## 4. Conclusions

According to the proposed study, it introduces a plant disease and infection detection method based on transformers that outperform existing SOTA studies. Additionally, to enhance the performance of the method, the proposed *GreenViT* was fine-tuned to bring down the number of parameters from 86 M to around 21.65 M. A total of three datasets, namely, the PV, DRLI and PC datasets, were employed to evaluate the proposed *GreenViT*. The study also showcases a comprehensive quantitative and qualitative analysis to prove the model generalization ability in real-world scenarios. In order to validate the efficacy and efficiency of the proposed approach, future experiments will utilize edge devices or drones that utilize a variety of leaf diseases. In the context of intelligent edge devices, the application of an attention-based model shows promise as a viable avenue of exploration that could be explored effectively.

## Figures and Tables

**Figure 1 sensors-23-06949-f001:**
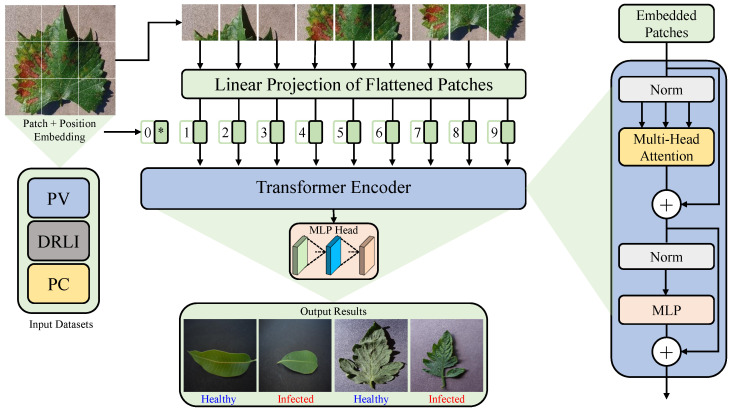
An overview of the proposed *GreenViT* framework for diagnosing plant diseases. The (*) notation in patch + position embedding represents class token, indicating that it depicts the information about the entire image in the sequence of patch embeddings.

**Figure 2 sensors-23-06949-f002:**
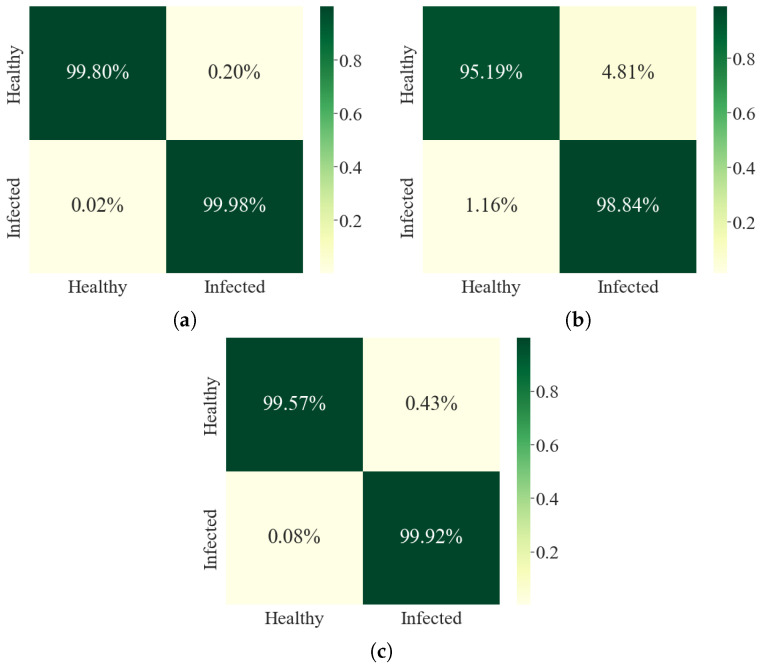
Confusion matrices of the proposed *GreenViT* for all the included datasets. (**a**) PlantVillage. (**b**) Data Repository of Leaf Images. (**c**) Plant Composite.

**Figure 3 sensors-23-06949-f003:**
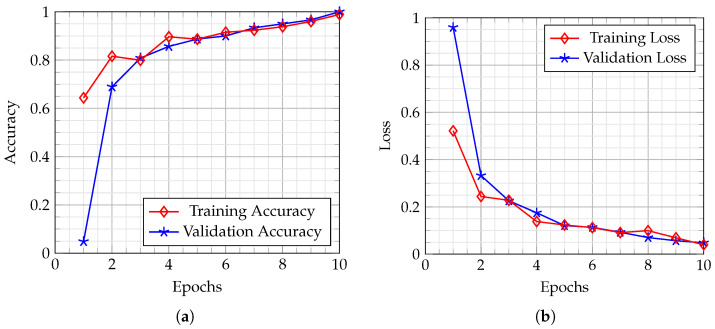
Training and validation accuracy and loss of the proposed *GreenViT* method on PC dataset. (**a**) Accuracy. (**b**) Loss.

**Figure 4 sensors-23-06949-f004:**
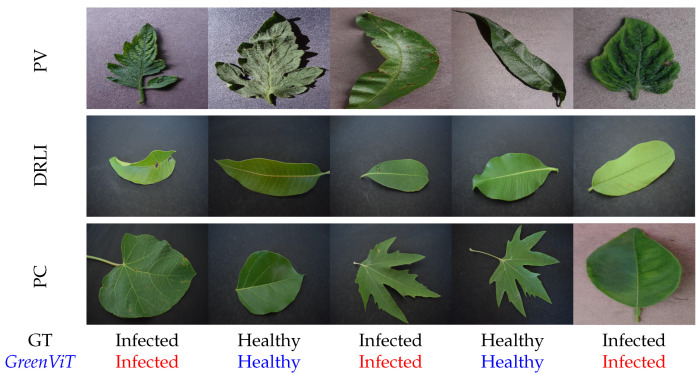
The objective of plant disease detection is to identify the presence of infections, but certain unique leaf images present challenges to the human eye and are not easily distinguishable without assistance. In our study, we addressed this issue by utilizing *GreenViT* to visually compare its performance on various datasets. The included figure displays a series of sample leaf images from different datasets. The first row consists of images from the PV, the second row showcases images from the DRLI and the third row represents images from the PC dataset. The second last row contains ground truth (GT) labels, where healthy samples are highlighted in **blue** text, while infected samples are indicated in **red** text. Through this visual comparison, we evaluated the effectiveness of *GreenViT* in detecting plant diseases across different datasets.

**Table 1 sensors-23-06949-t001:** The statistics of all three included datasets.

S. No.	Dataset	Training	Testing	Validation	Total Images
1	PlantVillage [13]	39,100	10,861	4344	54,305
2	Data Repository of Leaf Images [14]	3241	901	360	4502
3	Plant Composite [39]	42,341	11,762	4704	58,807

**Table 2 sensors-23-06949-t002:** The three basic ViT architectures, namely ViT-Base, ViT-Large and ViT-Huge, can be compared based on their layer count, hidden size (D), number of attention heads and parameters. The proposed *GreenViT* model is highlighted in blue color.

Model	No. of Layers	Hidden Size (D)	Heads	Parameters (M)
ViT Base	12	768	12	86
ViT Large	24	1024	16	307
ViT Huge	32	1280	16	632
* **GreenViT** *	**8**	**768**	**4**	**21.65**

**Table 3 sensors-23-06949-t003:** Quantitative evaluation of *GreenViT* in contrast to SOTA models using the included datasets. The proposed *GreenViT* model is highlighted in blue. The upward arrow (↑) depicts higher value is better.

Model	Class	PV				DRLI				PC			
		P	R	F1	ACC ↑	P	R	F1	ACC ↑	P	R	F1	ACC ↑
VGG19 [41]	Healthy	1.00	0.95	0.98	0.99	0.96	0.97	0.97	0.97	0.99	0.96	0.97	0.98
	Infected	0.98	1.00	0.99		0.97	0.96	0.96		0.98	1.00	0.99	
VGG16 [41]	Healthy	0.99	0.99	0.99	1.00	0.98	0.94	0.96	0.96	0.99	0.99	0.99	0.99
	Infected	1.00	1.00	1.00		0.93	0.98	0.96		0.98	1.00	1.00	
EfficientNetB0 [42]	Healthy	1.00	1.00	1.00	1.00	0.99	0.83	0.90	0.89	1.00	0.97	0.98	0.99
	Infected	1.00	1.00	1.00		0.78	0.98	0.87		0.99	1.00	0.99	
MobileNetV1 [43]	Healthy	1.00	0.99	0.99	1.00	0.97	0.96	0.97	0.97	0.99	0.99	0.99	0.99
	Infected	1.00	1.00	1.00		0.96	0.97	0.96		1.00	1.00	1.00	
MobileNetV3Small [44]	Healthy	1.00	1.00	1.00	1.00	0.93	0.99	0.96	0.96	1.00	0.99	0.99	0.99
	Infected	1.00	1.00	1.00		0.99	0.93	0.95		0.99	1.00	1.00	
ViT Base [26]	Healthy	0.92	0.98	0.95	0.95	0.81	0.62	0.70	0.75	0.87	0.95	0.91	0.94
	Infected	0.98	0.92	0.95		0.71	0.86	0.78		0.98	0.94	0.96	
*GreenViT*	Healthy	1.00	1.00	0.99	1.00	0.97	0.95	0.96	0.98	0.98	1.00	0.99	0.99
	Infected	0.99	1.00	1.00		0.98	0.99	0.98		0.99	0.98	0.99	

**Table 4 sensors-23-06949-t004:** Five-fold cross validation accuracies of *GreenViT* for the included datasets.

Fold	Dataset		
	PV	DRLI	PC
1	0.9836	0.9314	0.9540
2	0.9749	0.9425	0.9377
3	0.9723	0.9425	0.9471
4	0.9611	0.9623	0.9632
5	0.9839	0.9447	0.9680
Average Test Accuracy	0.9752	0.9446	0.9540

**Table 5 sensors-23-06949-t005:** Ten-fold cross validation accuracies of *GreenViT* for the included datasets.

Fold	Dataset		
	PV	DRLI	PC
1	0.9837	0.9647	0.9665
2	0.9543	0.9736	0.9603
3	0.9795	0.9713	0.9401
4	0.9681	0.9802	0.9540
5	0.9696	0.8450	0.9552
6	0.9812	0.9669	0.9620
7	0.9791	0.9425	0.9590
8	0.9828	0.9669	0.9580
9	0.9716	0.9337	0.9666
10	0.9698	0.9004	0.9574
Average Test Accuracy	0.9740	0.9445	0.9579

**Table 6 sensors-23-06949-t006:** An assessment of the proposed *GreenViT* FPS against several other DL models. This analysis provides the relative performance of each model in terms of inference speed. The proposed *GreenViT* model is highlighted in blue. The downward arrow (↓) illustrates that smaller value is better while upward arrow (↑) depicts higher value is better.

Model	Parameters (M) ↓	Size (MB) ↓	FPS ↑	
			RPi 4B+	CPU
VGG19	200.25	229.0	0.47	9.49
VGG16	147.15	168.0	0.62	11.09
EfficientNetB0	4.05	46.9	2.69	19.74
MobileNetV1	3.23	37.1	8.23	22.96
MobileNetV3Small	1.53	18.0	7.43	27.94
Vit Base	86.00	345.0	0.21	19.83
*GreenViT*	21.65	247.0	0.34	22.19

## Data Availability

Publicly available datasets were analyzed in this study. Link to the PV: https://github.com/spMohanty/PlantVillage-Dataset and DRLI: https://data.mendeley.com/datasets/hb74ynkjcn/1 (accessed on 5 July 2023).

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
