# Peer review of "Visual Intelligence in Precision Agriculture: Exploring Plant Disease Detection via Efficient Vision Transformers"

_sensors, 2023, doi:10.3390/s23156949_

Round 1

Reviewer 1 Report

The article is devoted to solving an applied problem using artificial intelligence methods. The topic of the article is relevant. The structure of the article is not classical for MDPI (Introduction, Models and Methods, Experiments, Discussion, Conclusions). The level of English is acceptable. The article is easy to read. The quality of the figures is good. The article cites 46 sources, some of which are not relevant.

The following comments and recommendations can be formulated on the material of the article:

1. The transformer uses multi-headed attention in three different ways: - In the encoder-decoder attention layers, requests come from the previous decoder layer, and memory keys and values come from the output of the encoder. This allows each position in the decoder to refer to all positions in the output sequence. This mimics typical encoder-decoder attention mechanisms in sequence-to-sequence models; - The encoder contains layers of self-attention. In such a layer, all keys, values, and requests come from the same place, in this case, the output of the previous encoder layer. Each position in the encoder can refer to all positions in the previous encoder layer; - Similarly, the self-attention layers in the decoder allow each position in the decoder to refer to all positions in the decoder up to and including the current position. Here it is important to prevent the information from moving to the left in the decoder in order to preserve the autoregressive property. The authors chose a mixed style. Please specify the reason.

2. Almost any dataset has a limited variety and does not cover all situations in which the correct operation of the model is desirable. This is especially evident in the case of complex data such as images, texts and sound recordings. There may be spurious correlations in the data, allowing one to predict the response with good accuracy only on a given sample without a "complex" understanding of the image. Datasets in which spurious correlations are explicitly expressed are called biased. It seems that all datasets in CV and NLP tasks are biased to one degree or another. I ask you to analyze the datasets used in this context.

3. The situation when the distribution of data differs in training and application is called distributional shift. In other words, the data on which the model will be applied differs, on average, from those on which the model was trained and tested. In particular, we can talk about the emergence of new types of examples or a change in the ratio of the number of examples of different types. This is due to the fact that the training data is often not diverse enough and does not cover all the types of examples on which the correct operation of the model is desired (or cover them in the wrong ratio). The good work of the model in terms of data shift is called out-of-distribution generalization. I ask the authors to justify whether their model is such.

4. To train transformers, you need a huge amount of data. That is why authors often rely on shortcuts caused by insufficient diversity of the training data distribution and the presence of spurious correlations in it (this resembles data leakage). Such a model only imitates the solution of the problem, and therefore may cease to work correctly if the conditions change. Please prove that this is not the case.

-

Author Response

Please see the attachement.
Thank you

Warm Regards

Authors

Reviewer 2 Report

A very interesting and practical topic.

The examples in Figure 4 showing infected and healthy plants are too easy to distinguish. Can you provide some examples from the database that are not easily distinguishable? Otherwise, it might give the impression that these datasets are too easy to differentiate.

1. In this paper, the authors talk about a new method called "GreenViT" for detecting plant infections and diseases. They use something called Vision Transformers (ViTs) to do this. However, in the paper, they only mention whether the leaves are healthy or not. It would be better if the method they proposed could actually identify what specific disease the plant is suffering from, which aligns better with what was summarized in the abstract.

2. The authors introduced a new model called GreenVit, which is compared with the traditional model called Vit. The two models differ in the number of layers, hidden size, and the number of heads used. However, besides these differences, it's not clear what other distinctions exist between them. Could the authors provide a clearer explanation?

3. Figure 4 provides examples of classifying 'Infected' and 'Healthy.' The images are easily distinguishable by the naked eye of non-experts, making it challenging to showcase the value of the method proposed in this paper. The authors should present more difficult examples that are hard for even experts to differentiate, but that GeenVit handles well.

Author Response

Please see the attachement.
Thank you

Warm Regards,

Authors

Round 2

Reviewer 1 Report

I have formulated the following recommendations and comments to the basic version of the article:

1. The transformer uses multi-headed attention in three different ways: - In the encoder-decoder attention layers, requests come from the previous decoder layer, and memory keys and values come from the output of the encoder. This allows each position in the decoder to refer to all positions in the output sequence. This mimics typical encoder-decoder attention mechanisms in sequence-to-sequence models; - The encoder contains layers of self-attention. In such a layer, all keys, values, and requests come from the same place, in this case, the output of the previous encoder layer. Each position in the encoder can refer to all positions in the previous encoder layer; - Similarly, the self-attention layers in the decoder allow each position in the decoder to refer to all positions in the decoder up to and including the current position. Here it is important to prevent the information from moving to the left in the decoder in order to preserve the autoregressive property. The authors chose a mixed style. Please specify the reason.

2. Almost any dataset has a limited variety and does not cover all situations in which the correct operation of the model is desirable. This is especially evident in the case of complex data such as images, texts and sound recordings. There may be spurious correlations in the data, allowing one to predict the response with good accuracy only on a given sample without a "complex" understanding of the image. Datasets in which spurious correlations are explicitly expressed are called biased. It seems that all datasets in CV and NLP tasks are biased to one degree or another. I ask you to analyze the datasets used in this context.

3. The situation when the distribution of data differs in training and application is called distributional shift. In other words, the data on which the model will be applied differs, on average, from those on which the model was trained and tested. In particular, we can talk about the emergence of new types of examples or a change in the ratio of the number of examples of different types. This is due to the fact that the training data is often not diverse enough and does not cover all the types of examples on which the correct operation of the model is desired (or cover them in the wrong ratio). The good work of the model in terms of data shift is called out-of-distribution generalization. I ask the authors to justify whether their model is such.

4. To train transformers, you need a huge amount of data. That is why authors often rely on shortcuts caused by insufficient diversity of the training data distribution and the presence of spurious correlations in it (this resembles data leakage). Such a model only imitates the solution of the problem, and therefore may cease to work correctly if the conditions change. Please prove that this is not the case.

The authors answered all of them. I liked the full and informative answers of the authors. I support the publication of the current version of the article. I wish the authors creative success.

-